# All-Trans Retinoic Acid Effect on *Candida albicans* Growth and Biofilm Formation

**DOI:** 10.3390/jof8101049

**Published:** 2022-10-05

**Authors:** Enrico Salvatore Pistoia, Terenzio Cosio, Elena Campione, Francesca Pica, Antonio Volpe, Daniele Marino, Paolo Di Francesco, Claudia Monari, Carla Fontana, Marco Favaro, Paola Zampini, Augusto Orlandi, Roberta Gaziano

**Affiliations:** 1Department of Experimental Medicine, University of Rome Tor Vergata, 00133 Rome, Italy; 2PhD Course in Microbiology, Immunology, Infectious Diseases, and Transplants (MIMIT), Department of Experimental Medicine, University of Rome Tor Vergata, 00133 Rome, Italy; 3Dermatology Unit, Department of Systems Medicine, Tor Vergata University Hospital, 00133 Rome, Italy; 4Anatomic Pathology, Department of Biomedicine and Prevention, University of Rome “Tor Vergata”, 00133 Rome, Italy; 5Department of Medicine and Surgery, Medical Microbiology Section, University of Perugia, 06129 Perugia, Italy; 6National Institute for Infectious Diseases (INMI) L. Spallanzani, IRCCS, 00149 Rome, Italy

**Keywords:** all-*trans* retinoic acid, *C. albicans*, morphotype switching, fungal infections, antifungal effect, anti-biofilm activity, TEM analysis

## Abstract

*Candida albicans (C. albicans)* is the most common fungal pathogen causing recurrent mucosal and life-threatening systemic infections. The ability to switch from yeast to hyphae and produce biofilm are the key virulence determinants of this fungus. In fact, *Candida* biofilms on medical devices represent the major risk factor for nosocomial bloodstream infections. Novel antifungal strategies are required given the severity of systemic candidiasis, especially in immunocompromised patients, and the lack of effective anti-biofilm treatments. Retinoids have gained attention recently due to their antifungal properties. Material and methods: The present study aimed at evaluating the in vitro effects of different concentrations (300 to 18.75 µg/mL) of All*-trans* Retinoic Acid (ATRA), a vitamin A metabolite, on *Candida* growth and biofilm formation. Results: ATRA completely inhibited the fungal growth, by acting as both fungicidal (at 300 µg/mL) and fungistatic (at 150 µg/mL) agent. Furthermore, ATRA was found to negatively affect *Candida* biofilm formation in terms of biomass, metabolic activity and morphology, in a dose-dependent manner, and intriguingly, its efficacy was as that of amphotericin B (AmB) (2–0.12 μg/mL). Additionally, transmission electron microscopy (TEM) analysis showed that at 300 μg/mL ATRA induced plasma membrane damage in *Candida* cells, confirming its direct toxic effect against the fungus. Conclusion: Altogether, the results suggest that ATRA has a potential for novel antifungal strategies aimed at preventing and controlling biofilm-associated *Candida* infections.

## 1. Introduction

Over the past two decades, invasive fungal infections (IFIs) caused by opportunistic fungi have emerged as a major cause of morbidity and mortality, especially in severely immunocompromised hosts, despite the availability of novel antifungal drugs in recent years [1,2]. Among *Candida* genus, *C. albicans* has been considered the most pathogenic species for its ability to switch between yeast to hyphal morphotype that is usually associated with invasion and tissue damage, and remains the most common cause of candidemia [3,4]. In the yeast form, *C. albicans* normally colonizes human skin as part of the cutaneous microbiota and mucosal surfaces [5]. Under conditions of immune dysfunction, *C. albicans* can become an opportunistic pathogen, causing recurrent mucosal and systemic infections with high mortality rates. *Candida* bloodstream infections (BSIs) are the third to the fourth most common cause of healthcare-associated infections [6]. However, the number of non-*albicans* species, including *C. parapsilosis*, *C. tropicalis, C. glabrata, C. lusitaniae, C. krusei* and more recently *C. auris*, a new fluconazole-resistant emerging species, has increased over recent decades, due to new invasive diagnostic and therapeutic procedures, other than the increase in the use of broad-spectrum antibiotics [7,8]. Besides its ability to switch from yeast to hyphal form, which represents the key virulence determinant, the formation of biofilms also plays a crucial role in the pathogenicity of *C. albicans* [9,10].

Biofilms are defined as microbial communities attached to a mucosal or basal surface and encased in an extracellular matrix that is composed of extracellular polymeric substances (EPS) such as polysaccharides, proteins and lipids [11,12]. Unlike other *Candida* spp., *C. albicans* biofilms display a more heterogeneous organization, composed by yeast cells, organized in a multilayer structure, pseudo-hyphae and true hyphae, surrounded by an extracellular matrix [12]. Mature biofilms continuously release *Candida* cells into the environment; these then colonize new sites, thus leading to chronic infections [13]. Furthermore, fungal cells embedded into biofilm display phenotypes that are distinct from their planktonic counterparts, including lower growth rates, especially those within deeper layers of mature biofilm, and much greater resistance to conventional antifungal agents, as well as to host immune response [14].

*C. albicans* species is the major cause of biofilms on medical devices (*e.g*., intravascular and urinary catheters, prostheses, vascular bypass grafts), which constitute the major risk factor for nosocomial bloodstream and deep tissue infections, representing approximately 50% of all nosocomial *Candida* infectious diseases [15,16,17,18]. It has been estimated that the mortality rate caused by biofilm-related *Candida* infections was almost double compared to planktonic infections [18]. *Candida* biofilms have also been found on vaginal mucosa surface in women suffering from recurrent vulvovaginal candidiasis (RVVC), thus acting as a persistent source of infection [19,20]. Treatments of *C. albicans* biofilm-associated infections are limited since the biofilm extracellular matrix acts as a physical barrier, thus preventing the diffusion of antimycotic drugs [10,21]. Because of the emergence of both superficial and systemic *Candida* biofilm-related infections, along with serious side-effects of current antifungal drugs, there is an urgent need to develop adequate and more effective therapeutic strategies capable of preventing biofilm formation or disrupting pre-formed biofilms.

In this scenario, retinoids may be promising candidates due to their remarkable antifungal activity [22]. The term “retinoid” refers to both natural and synthetic analogues of vitamin A. Retinoids can be classified into four generations, depending on their molecular structures, properties and their clinical applications in inflammatory, dyskeratotic, onco-hematological and infectious diseases [23,24,25]. Campione et al. demonstrated for the first time the therapeutic efficacy of tazarotene, a synthetic derivate of vitamin A, against dermatophytes both in vitro and in vivo in patients affected by onychomycosis [26]. ATRA, an active metabolite of vitamin A, has also been shown to suppress the germination of *C. albicans* and *A. fumigatus,* and synergize with the antifungals amphotericin B and posaconazole against *Aspergillus* growth in vitro, as well as to exert a protective effect in a rat model of invasive pulmonary aspergillosis (IPA) [27,28]. 

Considering that yeast germination and filamentous growth as pseudo-hyphae or true hyphae in *C. albicans* are crucial for biofilm formation and maintenance, the present work aimed at evaluating the in vitro antifungal and antibiofilm efficacy of ATRA on *C. albicans*, as well as its impact on ultrastructural features of *Candida* biofilm cells by TEM analysis. For comparative purposes and validation of the experiments, amphotericin B, one of the first-line agents for treating life-threatening invasive candidiasis, was also included in this study.

## 2. Materials and Methods

### 2.1. Candida Strain and Growth Conditions

All in vitro studies were carried out using *C. albicans* reference strain ATCC 20191. The strain was grown on Sabouraud dextrose agar (Difco Laboratories, Detroit, MI, USA), supplemented with chloramphenicol, for 24 h at 30 °C. After incubation, *Candida* yeasts were harvested by washing the slant culture with sterile saline. The cell density of *Candida* suspension was estimated by direct cell count, using a Bürker chamber, and adjusted to the desired concentration.

### 2.2. Antimicrobial Agents

Stock solutions of ATRA (catalog no. R2625; Sigma-Aldrich, Milan, Italy) and AmB (analytical grade powder; Sigma Aldrich, Milan, Italy) were dissolved in 50% dimethyl sulfoxide (DMSO; Sigma-Aldrich, Milan, Italy) and then with the appropriate test medium RPMI 1640 at a final concentration of 2.5% DMSO (*v*/*v*). DMSO at 2.5% was used for each experimental point in all assays. ATRA and AmB were tested at different concentrations ranging from 300 to 18.75 µg/mL and from 2 to 0.12 μg/mL, respectively, based on our previous works [28].

### 2.3. Antifungal Susceptibility Testing

The antifungal activity of ATRA against *C. albicans* was assessed using the broth microdilution method, as described in M27-A3-2017, a document produced by the Clinical and Laboratory Standards Institute (CLSI) [29] with some modifications. Briefly, RPMI 1640 with 2% glucose and L-glutamine (Sigma-Aldrich, Milan, Italy), buffered to pH 7.0, was used as fungal culture medium. A total of 100 μL of *Candida* suspension (2 × 10^6^/mL) were added to 96-well flat-bottom plates. Twofold serial dilutions of ATRA were prepared to final concentrations ranging from to 18.75 µg/mL, while AmB, ranging from 2 to 0.12 μg/mL, was chosen as the positive control drug in the study. Aliquots of 100 μL of ATRA or AmB were dispensed into each well. Positive (100 μL of culture medium plus 100 μL of *Candida*) and negative controls (200 μL of culture medium alone or 100 μL of culture medium plus 100 μL of each compound in the absence of *Candida*) were included in all experiments. The plates were then incubated with agitation (200 rpm) at 30 °C for 24 h. The minimum inhibitory concentrations (MICs) were determined spectrophotometrically at 510 nm, by using an enzyme-linked immunosorbent assay (ELISA) reader, in three independent experiments, carried out in triplicate. The MIC_90_ and MIC_50_ were defined as the lowest concentration of the compounds capable to inhibit 90% and 50% of fungal growth, respectively, compared to the drug-free control.

### 2.4. Biofilm Quantification by Crystal Violet and XTT Assays

In order to assess the effect of ATRA on *C. albicans* biofilm formation, in terms of total biomass and metabolic activity, 2 × 10^5^
*Candida* cells were cultured in 96-well plates in 200 μL of RPMI 1640 medium, supplemented with 10% fetal calf serum (FCS; Sigma-Aldrich, Milan, Italy) and incubated at 37 °C for 24 h in the absence or presence of different concentrations of ATRA or AmB, as described above. To evaluate the impact of ATRA on *Candida* pre-formed biofilm, 24 h-old biofilms were treated with ATRA and incubated for additional 24 h. Biofilm biomass was analyzed using crystal violet (CV) staining, as previously described [30,31]. Briefly, after incubation, each well was washed twice with Phosphate-buffered saline (PBS) solution (200 μL) and the plate was dried for 20 min at 35 °C. The washed biofilms were then stained with 150 µL of 0.4% aqueous CV solution for 20 min. After CV staining, the wells were washed three times with 200 µL of distillated water and CV bound to the biofilm was solubilized by adding 200 µL of 33% glacial acetic acid. After 15 min, 100 µL of destaining solution from each sample were transferred to a new plate and measured by a spectrophotometer plate reader at 595 nm. Three experiments in triplicate were carried out and the data were expressed as the arithmetic mean of absorbance values. 

Biofilm metabolic activity was assessed by an XTT [2,3-bis(2-methoxy-4-nitro-5-sulfophenyl)-2H-tetrazolium-5-carboxanilide sodium salt] reduction assay, according to established procedures [32], by mixing freshly prepared XTT and menadione solutions (Sigma-Aldrich, Milan, Italy) at 20∶1 (v/v). XTT-menadione solution (42 μL) and PBS (158 μL) were added to pre-washed biofilms and incubated at 37 °C in the dark for 3 h. After incubation, the obtained colored supernatant (100 μL) was transferred to a new microtiter plate and measured by absorbances at 490 nm. Three experiments in triplicate were performed and we then reported the arithmetic mean of absorbance values. In all experiments, the absorbance values of the negative control wells (containing no cells) were subtracted from the values of the test wells to account for any background absorbance.

### 2.5. Hyphal Growth Inhibition Assay

To evaluate the impact of ATRA on *Candida* germination and hyphal growth, 2 × 10^5^ cells were cultured in 96-well plates in 200 μL of RPMI 1640 medium, supplemented with 10% FCS, and incubated at 37 °C for 24 h in the absence or presence of different concentrations of ATRA or AmB. After incubation, the non-adherent cells were removed by washing the wells three times with PBS and crystal violet staining was employed to visualize microscopically the hyphal growth and biofilm morphology by using a light microscope (Olympus, Carl Zeiss, UK) with 40× magnification objective lenses. The images were documented with the accompanied digital camera.

### 2.6. Visualization and Quantification of Vitality of Candida albicans Cells

In order to assess the impact of ATRA on *C. albicans* viability, planktonic cells were double- stained with Calcofluor White (CW) and propidium iodide (PI) (Sigma-Aldrich, Milan, Italy). CW is a fluorescent blue dye that binds to cell-wall chitin of fungal cells, regardless of their metabolic state, while PI can enter only dead cells, generating red-fluorescence. Briefly, 200 μL of *C. albicans* cell suspension (1 × 10^6^ cells/mL), were incubated for 24 h at 30 °C, under continuous agitation at 200 rpm, in the absence or presence of ATRA in RPMI 1640 medium, supplemented with 10% FCS. After incubation, the cells were spotted onto slides and double-stained with CW (5 μg/μL) and PI (1 μg/mL), for 20 min in the dark at RT. Images were taken by using a fluorescent microscope (Olympus, Carl Zeiss, United Kingdom) at a magnification of 50×.

### 2.7. Observations by Transmission Electron Microscopy 

The effects of ATRA on *C. albicans* ultrastructural features and biofilm formation were also analyzed by TEM. To this end, *C. albicans* yeast cells (6 × 10^6^) were plated in a 6-well cell culture plate in 3 mL of RPMI 1640 medium, supplemented with 10% FCS, and incubated in the absence or presence of ATRA for 24 h at 37 °C. After incubation, the samples were fixed in Karnovsky’s solution, processed and embedded in EPON 812 [33]. These sections were stained with 0.1% toluidine blue, and ultrathin sections, counterstained with uranyl acetate and lead citrate, and photographed with H-7100FA Hitachi transmission electron microscope (Japan) at magnifications ranging from 2.5 k to 100 k.

### 2.8. Statistical Analysis

Data have been summarized as the means ± SD of three independent experiments performed in triplicate. Means were compared by using the Student’s *t*-test. The significance level for *p* values was considered as * *p* < 0.05; ** *p* < 0.01; *** *p* < 0.001.

## 3. Results

### 3.1. Antifungal Effect of ATRA against C. albicans Growth

In the present study, the antifungal activity of ATRA against *C. albicans* was evaluated by using the microdilution broth method and expressed as MIC values. The conventional antifungal agent AmB was used in the tests as a control drug. The data show a remarkable dose-dependent inhibitory effect of ATRA against the fungal growth, with MIC_90_ and MIC_50_ values of 150 μg/mL and 75 μg/mL, respectively, while the MIC_90_ and MIC_50_ for AmB were 1 μg/mL and 0.25 μg/mL, respectively. Based on these results *C. albicans* reference strain ATCC 20191, used in our experiments, was considered as susceptible to AmB in accordance with CLSI and European Committee for Antimicrobial Susceptibility Testing (EUCAST) breakpoints (susceptible MIC to AmB ≤ 1 μg/mL). The results suggest that ATRA at the concentrations ranging from 150 to 75 μg/mL exerts a significant inhibitory effect on *C. albicans* growth.

### 3.2. Effect of ATRA on C. albicans Biofilm Production 

*Candida* biofilm production was evaluated in terms of total biomass, by crystal violet staining, and metabolic activity, using the colorimetric XTT reduction assay, as described above. Data in Figure 1A show that ATRA at the highest concentrations (300–150 μg/mL) was able to completely arrest the biofilm production by *C. albicans* as AmB (2–1 μg/mL) (Figure 1B). In fact, no significant differences were observed between the mean OD values of *Candida* biomass after 24 h exposure to ATRA or AmB, and those of the corresponding cells at baseline (Appendix A). Doses of ATRA lower than 150 μg/mL (75–37.5 μg/mL), although they did not totally suppress the biofilm formation, induced a significant dose-dependent inhibition in biofilm development, similar to that of AmB (0.5–0.12 μg/mL). At concentrations lower than 37.5 μg/mL no significant differences were found between ATRA and the untreated control. 

These findings are in line with the metabolic activity of *Candida* biofilm, which was dose-dependently reduced by ATRA treatment (300–37.5 μg/mL), compared to the untreated control (Figure 2A). Similar results were obtained with AmB (2–0.12 μg/mL) (Figure 2B; Appendix A). Conversely, ATRA was not effective to inhibit *Candida* mature biofilm, even at high concentrations of 300 and 150 μg/mL (Appendix A). Altogether, our data suggest that ATRA exerts not only an inhibitory activity against *C. albicans* growth but also on biofilm development, by working as an antifungal agent, although these biological effects were reached at much higher doses than those of the conventional antifungal drug AmB.

### 3.3. ATRA Exerts a Dose-Dependent Fungicidal and/or Fungistatic Activity against C. albicans 

To determine whether the inhibitory effect of ATRA on *Candida* growth and biofilm formation was due to a fungistatic or fungicidal activity, the vitality of *C. albicans* planktonic cells was assessed by an immunofluorescence double staining with CW and PI. Calcofluor White stains the cell wall of fungi blue, regardless of the metabolic state of the fungal cells, while PI emits red fluorescence upon binding to DNA of dead cells. The microscopic images in Figure 3 show that exposure to ATRA 300 μg/mL resulted in a 60% of PI positive non-viable cells. By contrast, ATRA concentrations ranging from 150 to 37.5 μg/mL reduced the percentage of PI positive cells to less than 2%, i.e., a value very similar to that of control cells. The evidence that ATRA totally inhibited *Candida* growth and biofilm formation, without inducing cell death, suggests that this molecule has a great antifungal potential, by acting either as a fungicidal or as a fungistatic agent against *C. albicans,* depending on the concentration used.

### 3.4. Impact of ATRA on Yeast-to-Hyphal Dimorphic Transition of C. albicans

*C. albicans* switching from yeast-to-hyphal form is a critical step in the biofilm development. The hyphae within biofilms strongly contribute to biofilm stability by acting as a support scaffold for yeast cells and other hyphae. Herein, the impact of ATRA on *C. albicans* morphological transition from yeast to hyphal growth was microscopically evaluated with CV staining. The microscopic images reported in Figure 4 show that *Candida* biofilm in the control group was highly structured and composed of yeasts, pseudo-hyphae, and crisscrossing true hyphae. By contrast, *Candida* cells, when exposed to ATRA concentrations of 300 to 150 μg/mL, were unable to develop germ tubes and grow as filamentous forms, similar to those treated with AmB at concentrations ranging from 2 to 1 μg/mL. The morphological analyses also show that cell densities were significantly reduced in the presence of ATRA 300 μg/mL and 150 μg/mL, further reinforcing the evidence that high ATRA concentrations completely arrest biofilm formation by *C. albicans*. On the contrary, *Candida* yeast germination was not affected by low concentrations of ATRA (75–37.5 μg/mL), although the cells exhibited a delay in germ tube and hyphae development. Furthermore, the amount of elongated cell formation appeared significantly reduced when compared to the control group, while the proportion of cells with a round yeast shape increased. These effects were similar to those achieved with AmB at concentrations ranging from 0.5 to 0.25 μg/mL.

### 3.5. Ultrastructural Analysis of C. albicans Biofilm Cells upon Exposure to ATRA 

The ultrastructural features of *Candida* biofilm cells were further investigated by TEM analysis in the absence or presence of ATRA. As depicted in Figure 5, *C. albicans* control cells were arranged in small clusters, surrounded by an extracellular granular material, presumably resembling the extracellular polymeric matrix. Furthermore, the cells were metabolically active, capable of growing as budding yeasts or true hyphae and the cytosol was highly organized, as proven by the abundant number of small intracytoplasmic vacuoles and bilayered membranous extra-vesicles (EV). By contrast, as reported in Figure 6, *Candida* cells treated with ATRA (300 or 150 μg/mL) appeared isolated and unable to grow and release granular material. Evident signs of irreversible cell damage, as proven by discontinuity of the cytoplasmic membrane, were also observed in *Candida* cells treated with ATRA 300 μg/mL. Conversely, 37.5 μg/mL ATRA did not affect the ability of *Candida* to replicate and release extracellular granules. Overall, TEM images are in line with the previous results, confirming that ATRA at 300 μg/mL was able to exert a robust fungicidal activity, by damaging the integrity of the fungal cell membrane. In comparison, ATRA 150 μg/mL displayed a fungistatic effect, by inhibiting not only the cell replication and hyphal growth but also the secretion of extracellular material in *Candida* cells, without inducing cell damage. 

## 4. Discussion

Invasive fungal infections represent an emerging global health concern with invasive candidiasis and candidemia responsible for most cases [34,35,36]. At present, the treatment of *Candida* biofilm-related infections represents a critical medical problem due to the drug resistance of *Candida* cells when encased within the biofilm extracellular matrix. Various treatment options have been proposed for the prevention of biofilm formation, including antifungal lock therapy, employment of modified materials or coated surfaces, to prevent *Candida* yeast adherence and, consequently, biofilm development on medical devices, as well as to target the quorum sensing molecules involved in biofilm formation [37,38,39,40]. However, given the absence of a real solution, new fields have been explored in order to find new natural products or synthetic peptides that might have an effective action against *Candida* biofilms. In this scenario, retinoids might represent very promising molecules due to their antifungal activity both in human and experimental models [22]. The aim of this study was to assess the in vitro antifungal and anti-biofilm activity of ATRA, a naturally occurring derivative of vitamin A, under the family of retinoids, against the fungus *C. albicans*. 

Our results show that 300 μg/mL and 150 μg/mL ATRA concentrations completely arrested the growth of *C. albicans*, by exerting a double action on the fungus, i.e., being able to act either as a fungicidal or as a fungistatic agent in a concentration-dependent manner. In detail, the highest ATRA concentration tested (300 μg/mL), exhibited a clear-cut fungicidal activity, as attested by the high percentage of propidium iodide positive dead cells. This finding, supported by TEM analysis, is consistent with a disruption of the plasma membrane integrity, presumably due to the cell apoptosis or necrosis induced by ATRA treatment. The evidence that ATRA at this concentration exerted only a partial fungicidal effect resulting in the death of 60% of *Candida* cells, led us to hypothesize that its fungicidal activity might affect specific cell cycle phases of the fungus. Therefore, future flow cytometry studies are needed to investigate the impact of this retinoid on *C. albicans* cell cycle profile. On the other hand, the high proportion of living yeast cells observed after exposure to ATRA 150 μg/mL, showing an intact plasma membrane, impermeable to propidium iodide, but unable to replicate and grow as filamentous forms, suggests that ATRA also displayed a remarkable fungistatic activity against *C. albicans*. Furthermore, the results showed that ATRA negatively affected *Candida* biofilm formation in terms of total biomass, metabolic activity and morphology. Remarkably, ATRA at high concentrations (300 and 150 μg/mL) was able to completely prevent biofilm development as the antifungal drug AmB used at concentrations ranging from 2 μg/mL to 1 μg/mL. Although the antifungal efficacy was obtained with ATRA at much higher doses than those of AmB, it is very surprising how this natural compound may work as an antimycotic agent. By contrast, no effects were observed in vitro on pre-formed mature biofilm, not even at high doses of ATRA.

Biofilm formation in fungi is a complex process involving adhesion, hyphal growth, proliferation and production of a polymeric matrix. By means of the combination of its fungistatic and fungicidal activity, ATRA might prevent the biofilm production targeting critical steps during biofilm development. First of all, in *C. albicans* the hyphal growth plays a key role in the formation and maintenance of biofilms. In fact, within biofilms, hyphae form a skeleton that provides a support for yeast cells. Furthermore, a growing body of evidence indicates that *C. albicans* hypha-specific surface adhesins, i.e., ALS3, HWP1, EFG1, belonging to the agglutinin-like sequence (Als) family of proteins, play a central role in the biofilm production both on biotic and abiotic surfaces, by mediating *Candida* attachment to epithelial cells, endothelial cells, and extracellular matrix proteins [41]. Therefore, a conceivable explanation could be that ATRA is able to prevent the production of biofilm by inhibiting germ tube formation and hyphal growth in *Candida* yeasts, by acting not only on the biofilm structure but also on the hyphal-associated adhesin expression.

Moreover, hyphal morphogenesis in *C. albicans* is crucial for invasion and tissue damage. Notably, in the hyphal form *C. albicans* produces candidalysin, a cytolytic peptide toxin that is considered the critical destroying factor for epithelial cells at the mucosal surface, during *Candida* infection [42]. The ability to affect *C. albicans* hyphal growth further supports the therapeutic potential of ATRA in counteracting invasive *Candida* infections. In addition, differently from the pathogenetic hyphal forms, which are engulfed at a slower rate than yeast cells by macrophages and promote non-protective T helper (Th)-2 immune response, *Candida* yeast morphotypes are more susceptible to phagocytic cells and elicit Th1 protective immunity [43]. Thus, ATRA might be not only effective in preventing biofilm formation but might be used as a therapeutic regimen for the treatment of superficial or systemic biofilm-based *Candida* infections.

In second instance, it is known that microbial species are highly interactive during biofilm development and communicate via “*quorum-sensing*” (QS) molecules. In particular, farnesol is a key quorum sensing molecule of *C. albicans* and plays a central role in fungal growth, biofilm production and biofilm-associated antifungal drug resistance [44]. As with bacteria, in fungi one crucial feature for QS induction is the need for the pathogens to reach a critical cell population density, sufficient to overcome host immune response and establish the infection. Therefore, ATRA, by inhibiting *Candida* growth, may most likely prevent fungal cells from reaching the critical density threshold required to induce QS, and consequently arrest biofilm development.

Another key feature of biofilms is the production of an extracellular matrix by *C. albicans* cells during the maturation phase of biofilm development. This material encompasses the complex network of yeast, pseudo-hyphal and hyphal cells, providing a protective barrier to the surrounding environment and contributing to the architectural stability of the biofilm. Our results, obtained by TEM analysis, showed that the untreated control cells of *C. albicans* were metabolically active and released granular material into the extracellular space, which could presumably play a role in biofilm development as a component of the extracellular polymeric matrix [45]. This insight is supported by previous studies conducted by Chandra et al. on the development and architecture of *Candida* biofilm, in which the authors identified an amorphous extracellular granular material around *Candida* yeasts and hyphae as part of the biofilm extracellular matrix [46]. By contrast, *Candida* cells after exposure to high concentrations of ATRA appeared metabolically inactive and unable to release granular material into the extracellular space, suggesting that ATRA might also impair biofilm biomass development, by interfering with the extracellular matrix formation.

The molecular mechanisms underlying the antifungal and anti-biofilm activity of this retinoid still remain unclear. Several studies have identified the chaperone heat shock protein (HSP)-90 as one of the molecular regulators involved in the extracellular matrix production, during *C. albicans* biofilm growth [47]. A reduced expression of HSP90 led to a marked decrease of glucan content in the extra polymeric matrix, providing a convincing mechanism by which HSP90 might regulate in fungi the azole resistance associated with biofilms [47,48,49,50,51,52,53]. It has also been found that chaperone HSP90 governs biofilm dispersion. In this respect, in vitro studies demonstrated that compromising the HSP90 activity, induced a reduction of the biofilm production by *C. albicans* and impaired dispersal of biofilm cells, thus blocking their function as reservoirs that is essential for perpetuation of the fungal infection [47]. 

In our previous studies, in in silico models, we proposed that ATRA displayed a strong fungistatic effect against *Aspergillus fumigatus,* through its molecular interaction with the HSP90 ATP-binding site of *Aspergillus* conidia, thus compromising the function of the protein [28]. These findings led us to hypothesize that ATRA may impair biofilm extracellular matrix production via its molecular interaction with *C. albicans* HSP90. 

In addition to being a potent antifungal agent, ATRA was found to possess important immunomodulatory properties on both innate and adaptive immune response [54,55], which could represent an adjunctive value to its therapeutic potential in *Candida* infections, especially in immunocompromised hosts. Therefore, the potential employment of this molecule could be particularly beneficial in selected patients, such as individuals undergoing medical devices implantation, organ solid and bone marrow transplantation, aimed at preventing systemic mycoses, or in the prophylaxis of recurrent mucosal *Candida* infections such as oral or vulvovaginal candidiasis. The protective role of vitamin A against human pathogens such as bacteria, viruses, protozoa and fungi, including *C. albicans*, further reinforce the concept that retinoids have a great antimicrobial potential. In this respect, the serum levels of both vitamin A and its active metabolite, ATRA, have been proposed as a predictive biomarker for *Candida* infection development in psoriatic patients [25].

It should be pointed out that ATRA is currently used *per os* in clinical practice to treat patients affected by promyelocytic leukemia, due to its ability to differentiate leukemic promyelocytes rapidly [56]. ATRA has also been used to treat various forms of malignancies including, breast cancer and liver cancer [57,58,59]. However, endogenous serum concentrations of ATRA in healthy individuals are about 2 nM, much lower than those resulted to be effective against *C. albicans* in our experimental models, ranging from 300 to 37.5 μg/mL [60]. This may represent a limitation for the systemic use of this drug. To avoid systemic side effects, recently, novel formulation based on nanocarriers have been developed, such as ATRA-loaded nanomicelles (ATRA-TPGSs), obtained by its encapsulation in D-α-tocopheryl-polyethylene-glycol-succinate (TPGS). This drug delivery system offers the advantage of positioning the bioactive molecule in direct contact with the fungal-infected tissues, enhancing the drug bioavailability, and thus, ensuring a rapid onset of the therapeutic response [22]. Furthermore, more in-depth studies are needed to address properly all these issues before such compound can be translated in clinic for the prevention and/or treatment of fungal diseases.

## 5. Conclusions

Considering its multiple beneficial effects and its safety profile, this retinoid, either as local therapy or into coated medical devices with a presumably prolonged and higher drug release, minimizing side effects of systemic treatments, could be a promising option for preventing and/or treating biofilm-related *Candida* infections. Furthermore, ATRA could be administered as a monotherapy or in combination with low doses of conventional antifungal drugs, offering the advantages of limiting the major clinical hurdle of drug resistance in fungi and the side-effects related to currently available antifungal therapies.

Further studies are needed to understand the molecular mechanisms by which ATRA exerts the antifungal and anti-biofilm activity against *C. albicans*, as well as to investigate the impact of this retinoid on adhesin protein-encoding gene expression, i.e., *ALS3, HWP1*, and *ECE1*, which are involved in the adhesion process to host mucosal/prosthetic surfaces and biofilm formation both in vitro and in vivo.

A possible limitation of our research is related to the use of a single amphotericin B susceptible *C. albicans* reference strain in our experiments. Therefore, further work is needed, including drug-resistant clinical isolates of *C. albicans* or other *Candida* species, such as the emerging multi-drug resistant *C. auris,* to confirm definitely the antifungal efficacy of this retinoid.

## Figures and Tables

**Figure 1 jof-08-01049-f001:**
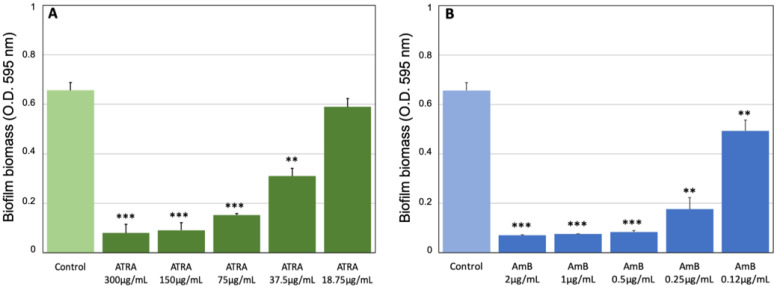
Effect of ATRA on biofilm formation by *C. albicans*. (**A**) *Candida* cultures were incubated for 24 h in the absence or presence of ATRA at various concentrations (300–18.75 μg/mL). Biofilm growth was analyzed as total biomass by crystal violet assay. (**B**) AmB (2–0.12 μg/mL) was used as positive control drug. The absorbance intensity of the crystal violet dye was measured using a spectrophotometer plate reader at 595 nm. Results are the means ± SD of three independent experiments carried out in triplicate. ** *p*< 0.01; *** *p*< 0.001.

**Figure 2 jof-08-01049-f002:**
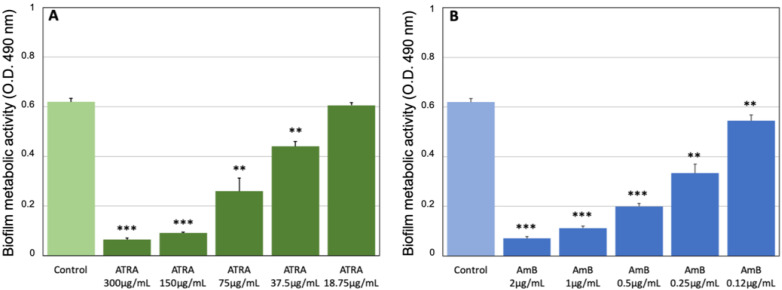
Effect of ATRA on metabolic activity of *C. albicans* biofilm. (**A**) *Candida* cultures were incubated for 24 h in the absence or presence of ATRA at various concentrations (300–18.75 μg/mL). (**B**) AmB (2–0.12 μg/mL) was used as positive control. Biofilm metabolic activity was analyzed by XTT assay, using a spectrophotometer plate reader at 490 nm. Results are the means ± SD of three independent experiments carried out in triplicate. ** *p*< 0.01; *** *p*< 0.001.

**Figure 3 jof-08-01049-f003:**
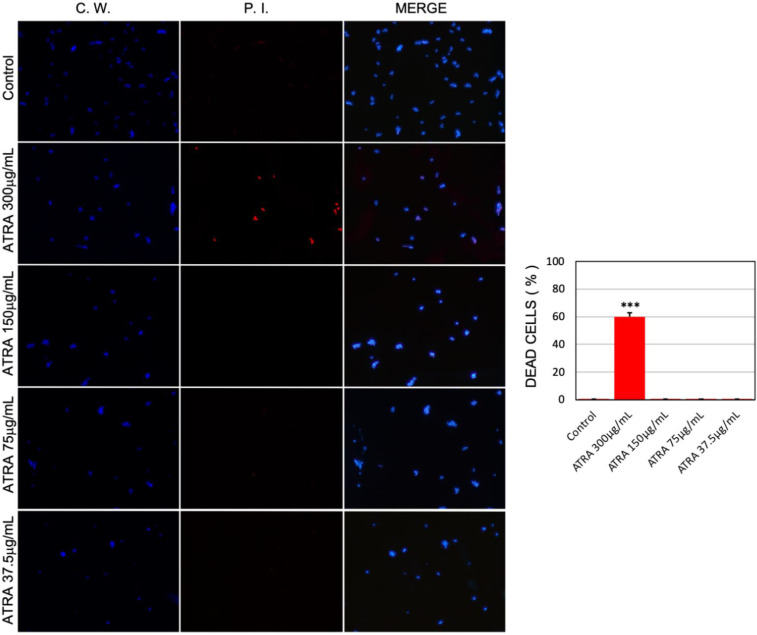
Visualization of *C. albicans* vitality by Calcofluor White (CW) and propidium iodide (PI) double staining. *C. albicans* planktonic cells were incubated without or with ATRA at various concentrations (300–37.5 μg/mL) for 24 h and then double-stained with CW and PI. Fluorescent images of the cells are shown. First column: Blue fluorescence represents the cell wall stained by CW; Second column: Red fluorescence represents PI-stained nucleic acids of dead cells; Last column: Merged images showing cells labeled with CW + PI. All images were obtained at 50× magnification by using a fluorescence microscope. The results are presented in histogram format as percentage of dead cells. At least 10 fields were counted per slide. *** *p* value <0.001.

**Figure 4 jof-08-01049-f004:**
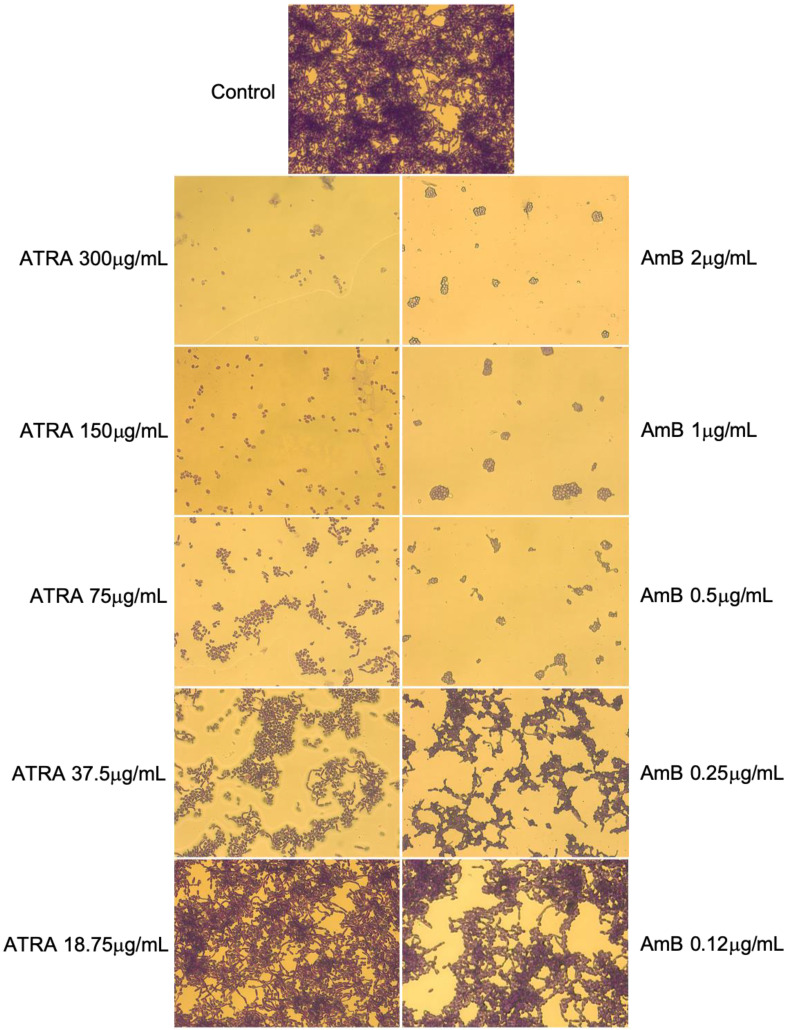
Effect of ATRA on phenotypic switching of *C. albicans*. *Candida* cultures were incubated for 24 h in the absence or presence of ATRA at different concentrations (300–18.75 μg/mL). The phenotypic transition from yeast-to-hyphal morphotype was analyzed by light microscopy after crystal violet staining. AmB was used as positive control drug. The images were acquired by a light microscope (Olympus, Carl Zeiss, UK) with 40× magnification objective lenses. One representative experiment of three is shown.

**Figure 5 jof-08-01049-f005:**
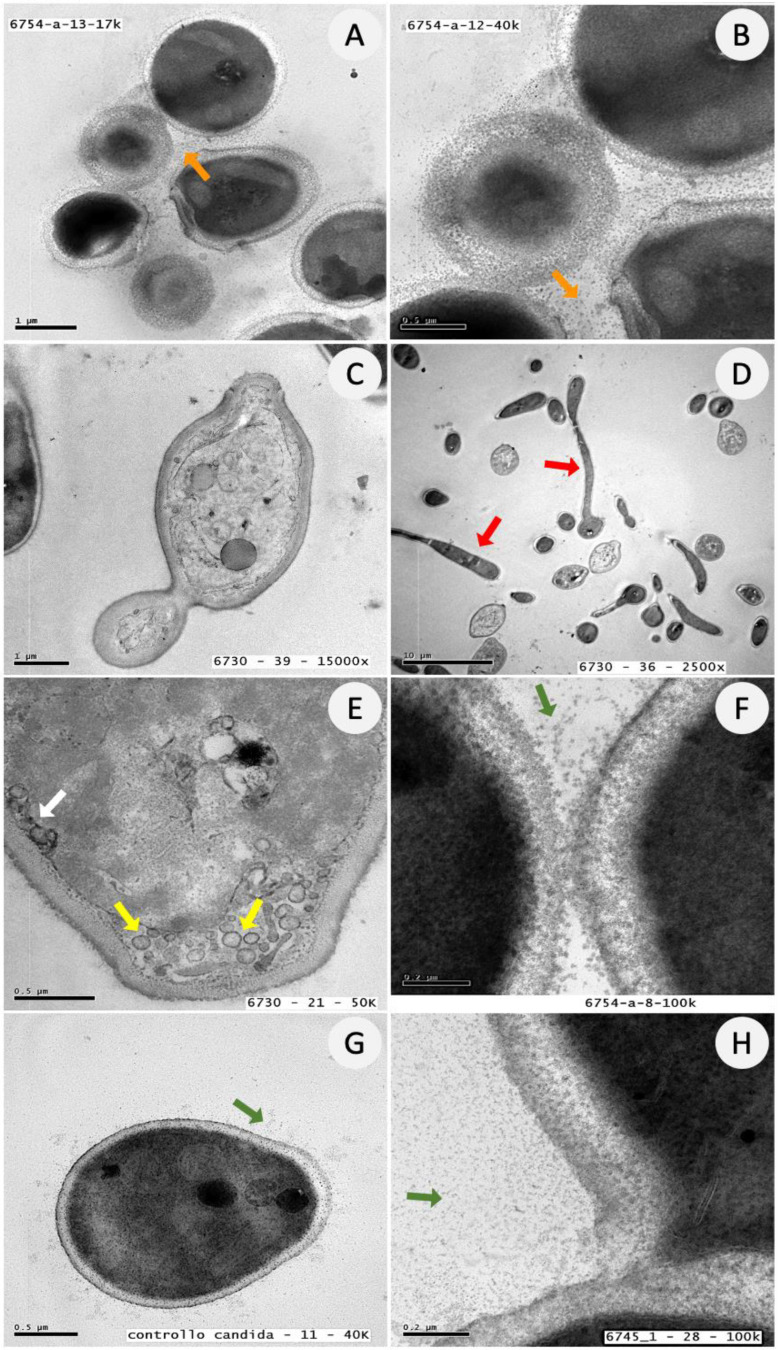
Ultrastructural features of *C. albicans* cells by TEM analysis. A-H: Representative images of *C. albicans* control cells. (**A**,**B**) cells arranged in small clusters and surrounded by extracellular granular material (orange arrows); (**C**) *Candida* replication in budding yeast; (**D**) budding yeasts and true hyphae (red arrows); (**E**) cells showing numerous intracytoplasmic vacuoles (yellow arrows) and EV (white arrow) at the periplasmic space; (**F**–**H**) cells secreting extracellular granular material (green arrows).

**Figure 6 jof-08-01049-f006:**
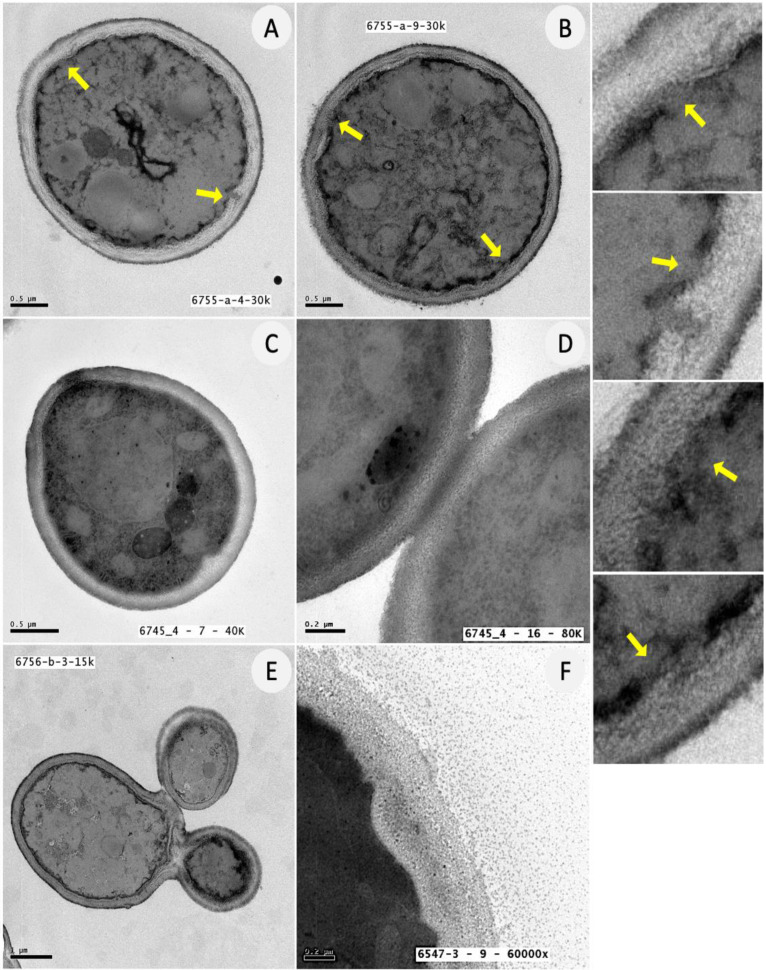
Ultrastructural features of *C. albicans* cells after ATRA treatment by TEM analysis. (**A**–**F**) Representative images of *C. albicans* cells treated with ATRA at different concentrations for 24 h. (**A**,**B**) ATRA 300 μg/mL; (**C**,**D**) ATRA 150 μg/mL; (**E**,**F**) ATRA 37.5 μg/mL. (**A**–**D**) TEM images show isolated and metabolically inactive *Candida* yeasts, incapable of replicating in budding daughter cells and growing in hyphal forms; no extracellular granular material was detected around *Candida* cells; (**A**,**B**) evidence of cell damage with discontinuities of the plasma membrane (yellow arrows); (**E**,**F**) *Candida* cells replicating as budding yeasts and releasing extracellular granules.

## Data Availability

The data presented in this study are available on request from the corresponding author.

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
