# Peer review of "All-Trans Retinoic Acid Effect on *Candida albicans* Growth and Biofilm Formation"

_jof, 2022, doi:10.3390/jof8101049_

Round 1
Reviewer 1 Report
In this very interesting manuscript on the effects of all-trans retinoic acid (ATRA)on Candida albicans growth and biofilm formation, the authors build on their previous reports of the fungistatic activity of ATRA (Campione et al, 2016). The experimental design of this new paper is however more comprehensive than their previous publication, addressing additional aspects of Candida growth control, such as the biofilm formation. The manuscript is very well written, the methodology is sound and the general outline is clear.
However, there are a few aspects in the both methodology and also the interpretation of the data that need to be clarified prior to publication.
Major comments:
1) It is not clear to me what controls were used in the study to address the specific effect of ATRA. The authors state that ATRA was diluted in DMSO, which is known to have an impact on Candida growth (by its own) at relatively low concentrations (Hazen et al, 2012; Akram Randhawa et al, 2008). It would be important to state the final concentration of DMSO in the growth medium (where ATRA was added), and whether the controls for the effect of ATRA were supplemented with equal quantities of the solvent (i.e. DMSO alone). Otherwise, it might be argued whether the observed effect is due to the DMSO, and not ATRA (especially since the concentrations of ATRA are pretty high....more on this in my second comment).
2) The concentrations of ATRA that were fungicidal or fungistatic are relatively high. The authors report a "potent" effect (line 285), but as everything, this is relative. The authors report effects on the growth of Candida in a concentration range between 0.12 and 1 mM. Furthermore, they obtained minimum inhibitory concentrations (MICs) at 0.5 mM (MIC90) and 0.25 mM (MIC50). In this same paragraph, they also report the MICs for AmB (which was used as positive control), and they conclude that "the antifungal effect of ATRA was comparable to that of AmB" (line 198). The concentration units reported for the MICs for both compounds (ATRA and AmB) were different, so I did the maths. The reported MIC90 for AmB was 1 µg/mL, which actually is around 1 µM. That means that concentration-wise the effect of AmB is 500x more potent than that measured for ATRA (same MIC90 is at 0.5 mM, i.e. 500 µM). If not concentration-wise, how did the authors measure the effectivity of the inhibition to state that the effect is comparable? It is not clear to me what exactly is comparable, especially in the paragraph reporting MICs. The same applies for further statements in line 2016 (“….comparable in terms of efficacy to that of the antifungal AmB”) and line 342 (“efficacy very 342 similar to that achieved by means of the antifungal drug AmB”). The relative magnitude of the inhibitory effect of ATRA (in comparison to AmB) should be clearly stated (this might include the use of only one universal concentration unit for both compounds).
3) The relative high effective dose of ATRA on Candida growth has a second consequence: the clinical interpretation of the data. The authors suggest its therapeutic use. This should be stated with caution, as the concentrations tested in this manuscript are far beyond any physiological concentrations of ATRA. Different studies report serum and plasma concentrations of ATRA in the nM (nanomolar) range. At such concentrations, ATRA already exerts several pleiotropic effects on human cells. The mM (milimolar) range is very far, and as stated by the authors, the inhibitory effect of ATRA on fungal cells is limited to concentrations above 0.12 mM. Thus, the limitations of the study and its interpretation must be clearly stated, especially regarding the high concentration of ATRA (when compared to physiological concentrations).
Minor comments:
4) line 70: It is not clear to me WHAT represents that “50% of all nosocomial infections”.
Author Response
Reviewer 1
In this very interesting manuscript on the effects of all-trans retinoic acid (ATRA) on Candida albicans growth and biofilm formation, the authors build on their previous reports of the fungistatic activity of ATRA (Campione et al, 2016). The experimental design of this new paper is however more comprehensive than their previous publication, addressing additional aspects of Candida growth control, such as the biofilm formation. The manuscript is very well written, the methodology is sound and the general outline is clear.
Dear Reviewer, thank you very much for your positive evaluation and comments on our manuscript. We have addressed all your concerns in the revised version of the manuscript. Please find below our detailed replies.
However, there are a few aspects in the both methodology and also the interpretation of the data that need to be clarified prior to publication.
Major comments:
- It is not clear to me what controls were used in the study to address the specific effect of ATRA. The authors state that ATRA was diluted in DMSO, which is known to have an impact on Candida growth (by its own) at relatively low concentrations (Hazen et al, 2012; Akram Randhawa et al, 2008). It would be important to state the final concentration of DMSO in the growth medium (where ATRA was added), and whether the controls for the effect of ATRA were supplemented with equal quantities of the solvent (i.e. DMSO alone). Otherwise, it might be argued whether the observed effect is due to the DMSO, and not ATRA (especially since the concentrations of ATRA are pretty high....more on this in my second comment).
Authors: We thank the Reviewer for pointing this out. DMSO has been employed at a final concentration of 2.5% v/v for each experimental point (including Candida control group) in all assays, as we specified in the Material and Methods section. At this concentration DMSO did not affect Candida growth and germination in our experimental models. In this respect, we have provided the images of Candida cultures in the absence or presence of DMSO, showing that DMSO at 2.5% did not inhibit germination of Candida yeasts. Regarding the impact of DMSO on Candida growth there are conflicting data in the literature. A dose-dependent inhibitory effect of DMSO (between 2.5 and 7.5%) on Candida germination was reported by Akram Randhawa et al. On the other hand, Schamberger B et al, demonstrated that DMSO even at high concentration (10% v/v) does not have toxic effect against C. albicans.
Akram Randhawa M. Dimethyl sulfoxide (DMSO) inhibits the germination of Candida albicans and the arthrospores of Trichophyton mentagrophytes. Nihon Ishinkin Gakkai Zasshi. 2008;49(2):125-8. doi: 10.3314/jjmm.49.125.
Schamberger B, Plaetzer K. Photofungizides Based on Curcumin and Derivates Thereof against Candida albicans and Aspergillus niger. Antibiotics (Basel). 2021 Oct 28;10(11):1315. doi: 10.3390/antibiotics10111315
- The concentrations of ATRA that were fungicidal or fungistatic are relatively high. The authors report a "potent" effect (line 285), but as everything, this is relative. The authors report effects on the growth of Candida in a concentration range between 0.12 and 1 mM. Furthermore, they obtained minimum inhibitory concentrations (MICs) at 0.5 mM (MIC90) and 0.25 mM (MIC50). In this same paragraph, they also report the MICs for AmB (which was used as positive control), and they conclude that "the antifungal effect of ATRA was comparable to that of AmB" (line 198). The concentration units reported for the MICs for both compounds (ATRA and AmB) were different, so I did the maths. The reported MIC90 for AmB was 1 µg/mL, which actually is around 1 µM. That means that concentration-wise the effect of AmB is 500x more potent than that measured for ATRA (same MIC90 is at 0.5 mM, i.e. 500 µM). If not concentration-wise, how did the authors measure the effectivity of the inhibition to state that the effect is comparable? It is not clear to me what exactly is comparable, especially in the paragraph reporting MICs. The same applies for further statements in line 2016 (“….comparable in terms of efficacy to that of the antifungal AmB”) and line 342 (“efficacy very 342 similar to that achieved by means of the antifungal drug AmB”). The relative magnitude of the inhibitory effect of ATRA (in comparison to AmB) should be clearly stated (this might include the use of only one universal concentration unit for both compounds).
Authors: We agree with the Reviewer that the fungicidal/fungistatic effect was obtained with high doses of ATRA ranging from 300-37.5 mg/mL, but it should be noted that ATRA is a natural compound and many natural substances display antimicrobial activities through concentration ranges in microgr/mL. Thus, as suggested, we have deleted the term “potent”. Furthermore, our intent was not to compare the antifungal efficacy of ATRA with that of AmB in terms of concentrations. Indeed, the MIC90 value of ATRA resulted to be 150 times higher than that of AmB. But we just wanted to stress the concept that a natural compound, such as ATRA, may exert an inhibitory effect against Candida growth and biofilm formation, working as an antifungal agent. For greater clarity we have deleted the following ambiguous sentence “The results indicated that the antifungal effect of ATRA was comparable to that of AmB, suggesting a very promising therapeutic potential of this retinoid“. Moreover, the sentence “Altogether, our data suggest that ATRA exerts a potent inhibitory activity against C. albicans growth and also displays a strong anti- biofilm effect comparable in terms of efficacy to that of the antifungal AmB” has been modified as follows: “Altogether, our data suggest that ATRA exerts not only an inhibitory activity against C. albicans growth but also on biofilm development, by working as an antifungal agent, although these biological effects were reached at much higher doses than those of the conventional antifungal drug AmB”. We have also converted the concentration unit of ATRA from molar to microgram as AmB, in order to use only one universal concentration for both compounds as suggested.
3) The relative high effective dose of ATRA on Candida growth has a second consequence: the clinical interpretation of the data. The authors suggest its therapeutic use. This should be stated with caution, as the concentrations tested in this manuscript are far beyond any physiological concentrations of ATRA. Different studies report serum and plasma concentrations of ATRA in the nM (nanomolar) range. At such concentrations, ATRA already exerts several pleiotropic effects on human cells. The mM (milimolar) range is very far, and as stated by the authors, the inhibitory effect of ATRA on fungal cells is limited to concentrations above 0.12 mM. Thus, the limitations of the study and its interpretation must be clearly stated, especially regarding the high concentration of ATRA (when compared to physiological concentrations).
Authors: Dear Reviewer, thank you very much for pointing out this aspect. ATRA is currently used per os in clinical practice to treat patients affected by promyelocytic leukemia, at a dose of 45 mg/m2 of body surface (divided in two equal administrations), corresponding to 80 mg per day [Girmenia et al, 2003]. However, endogenous serum concentrations of ATRA in healthy individuals are about 2 nM [Arnold et al 2012], much lower than those found to be effective against C. albicans in our experimental models, ranging from 300 to 37.5 mg/mL. This could represent a limitation for the systemic use of this drug. Today, clinical use of ATRA is quite limited indicating the need of additional of more in-depth studies. The data available in literature have been reported in the revised version of the manuscript where this point has been addressed and discussed (see, lines 449-459; 462-465).
Girmenia, C.; Lo Coco, F.; Breccia, M.; Latagliata, R.; Spadea, A.; D'Andrea, M.; Gentile, G.; Micozzi, A.; Alimena, G.; Martino, P.; Mandelli, F. Infectious complications in patients with acute promyelocytic leukaemia treated with the AIDA regimen. Leukemia. 2003, 17, 925-930. doi:10.1038/sj.leu.2402899.
Arnold SLM, Amory JK, Walsh TJ, Isoherranen N. A sensitive and specific method for measurement of multiple retinoids in human serum with UHPLC-MS/MS. J Lipid Res. 2012;53(3):587-598. doi:10.1194/jlr.D019745
Minor comments:
4) line 70: It is not clear to me WHAT represents that “50% of all nosocomial infections”.
4) Dear Reviewer, thank you for pointing this out. The sentence “50% of all nosocomial infections” seems to be misleading in this context. So, we have replaced it with “representing approximately 50% of all nosocomial Candida infectious diseases “.
Thank you and best regards

Reviewer 2 Report
This study evaluates the antifungal activity of retinoid atRA against Candida albicans in vitro. The authors show that at concentration of 1-0.5 mM, atRA inhibits planktonic growth of Candida as well as inhibits biofilm activity. This biofilm inhibition is supported by imaging as well as metabolic activity assays. The data presented in the study is promising. However, use of physiologically relevant models and strains as well as mechanistic insight is required to increase the impact of the study.
Comments:
1. The use of the word new in the title should be avoided. Rather the title should reflect the major takeaway point of the paper
2. The authors report results with one strain of C. albicans. More strains, and especially clinical isolates, should be included to judge the generality of the antifungal effects of atRA. Additionally evaluation of atRA against AmB-resistant strains would add significantly to the impact of the paper.
3. Can the authors clarify if appropriate vehicle controls were used for MIC and biofilm inhibition assays?
4. The authors effectively show that atRA inhibits biofilm formation by Candida in a dose-dependent manner. it would benefit the study greatly if the authors could do a time course of atRA treatment with Candida and quantify effects at different phases of biofilm formation along with gene expression analysis to look at the effects on biofilm formation genes such as adhesion proteins.
5. In the biofilm assays, since atRA treatment is at the same time as Candida inoculation, it is hard to determine if the fungicidal effect of atRA is responsible for lack of biofilm formation or whether it is a result of an effect on biofilm formation genes. It would be interesting to stagger the atRA treatment and Candida inoculation for e.g. 1h, 3h, 24h etc post Candida inoculation to see the outcome of atRA treatment.
6. A crucial attribute of any antifungal would be it’s ability to disrupt Candida biofilms on catheters and other implants. The authors mention in the discussion section that atRA did not show activity against preformed biofilms, however data is not shown. It is recommended that this data be added to the manuscript.
7. Have the authors evaluated the ability of atRA to synergize with AmB or other clinically relevant antifungal compounds? This is especially relevant given the rise of resistance in Candida spp.
8. The authors report the inhibitory effect of atRA on yeast to hyphal transition using images. It is unclear from the methods if this was done under hyphae inducing conditions used in other papers such as Sun et al 2015, Ha et al 1999 etc. The representative images also need to be supported by quantification to support the conclusions.
9. Given the potent effects of atRA on mammalian cells, the authors should discuss the existing knowledge regarding the use of atRA as a therapeutic agent, in terms of bioavailability, safety, potential cytotoxicity etc.

Author Response
Reviewer 2
This study evaluates the antifungal activity of retinoid atRA against Candida albicans in vitro. The authors show that at concentration of 1-0.5 mM, atRA inhibits planktonic growth of Candida as well as inhibits biofilm activity. This biofilm inhibition is supported by imaging as well as metabolic activity assays. The data presented in the study is promising. However, use of physiologically relevant models and strains as well as mechanistic insight is required to increase the impact of the study.
Authors: Dear Reviewer, thank you very much for your valuable comments. We agree with you that our study has some limitations, including missing information on the potential mechanisms of action of ATRA, the lack of physiologically relevant models, i.e. biofilm formation on catheter surfaces, as well as the use of only one AmB-sensitive Candida strain. However, this is the first evidence on the anti-biofilm activity of ATRA and it seemed interesting to us to publish these preliminary data. Future work is needed to investigate the molecular mechanisms related to the antifungal properties of this retinoid and its efficacy on clinical isolates of Candida as well as on Candida strains resistant to AmB.
Comments:
- The use of the word new in the title should be avoided. Rather the title should reflect the major takeaway point of the paper.
Authors: Dear Reviewer, we have modified the title as suggested and replaced it with “All-trans retinoic acid effect on Candida albicans growth and biofilm formation”.
- The authors report results with one strain of C. albicans. More strains, and especially clinical isolates, should be included to judge the generality of the antifungal effects of atRA. Additionally evaluation of atRA against AmB-resistant strains would add significantly to the impact of the paper.
Authors: Dear Reviewer, we agree that the use of clinical isolates, and also AmB-resistant Candida strains undoubtedly will improve the impact of our study. In this preliminary study we have preferred to use an ATCC reference strain of C. albicans (ATCC 20191). However, as we reported in the conclusion section, further studies will be carried out to investigate the antifungal efficacy of ATRA on clinical isolates of Candida albicans as well as on multidrug resistance Candida strains, such as Candida auris.
- Can the authors clarify if appropriate vehicle controls were used for MIC and biofilm inhibition assays?
Authors: Dear Reviewer, thank you to underline this point. DMSO was used as vehicle for both MIC and biofilm inhibition assay. Both ATRA and AmB were dissolved in 50% DMSO at a final concentration of 2.5% v/v for each experimental point (including Candida control group) in all assays, as we specified in the Material and Methods section. At this concentration DMSO did not affect Candida growth and germination in our experimental models. In this respect, we have provided the images (see attached) of Candida cultures in the absence or presence of DMSO, showing that DMSO at 2.5% did not inhibit germination of Candida yeasts.
CLSI. Reference Method for Broth Dilution Antifungal Susceptibility Testing of Yeasts. 4th ed. CLSI standard M27. Wayne, PA: Clinical and Laboratory Standards Institute; 2017
- The authors effectively show that atRA inhibits biofilm formation by Candida in a dose-dependent manner. it would benefit the study greatly if the authors could do a time course of atRA treatment with Candida and quantify effects at different phases of biofilm formation along with gene expression analysis to look at the effects on biofilm formation genes such as adhesion proteins.
Authors: Dear Reviewer, thank you for your suggestion. The rationale for using a 24-hour time point was based on preliminary data indicating that by 24 hours, Candida cells produced a mature, well-structured biofilm. As suggested by the reviewer it would be interesting in the future to perform a time course of ATRA treatment to evaluate the impact of this retinoid during the different phases of the biofilm formation, in relation to the expression of adhesin protein-encoding genes i.e.: ALS3, HWP1, and ECE1.
- In the biofilm assays, since atRA treatment is at the same time as Candida inoculation, it is hard to determine if the fungicidal effect of atRA is responsible for lack of biofilm formation or whether it is a result of an effect on biofilm formation genes. It would be interesting to stagger the atRA treatment and Candida inoculation for e.g. 1h, 3h, 24h etc post Candida inoculation to see the outcome of atRA treatment.
Authors: Dear Reviewer, thank you for your valuable comment. In this preliminary study, as we stated above, we have chosen only one time point. It would be interesting in the future to evaluate the effect of ATRA at different time points from Candida inoculation to better understand whether at 300 mg/mL the inhibitory effect of ATRA on the biofilm production is due to its fungicidal effect or its impact on the hyphal specific and biofilm-associated gene expression. At 150 mg/mL ATRA exerts only a fungistatic activity. Thus, we hypothesize that the lack of biofilm formation may be related to its ability to modulate the expression of hyphal and biofilm-associated genes.
- A crucial attribute of any antifungal would be it’s ability to disrupt Candida biofilms on catheters and other implants. The authors mention in the discussion section that atRA did not show activity against preformed biofilms, however data is not shown. It is recommended that this data be added to the manuscript.
Authors: Dear Reviewer, thank you for the suggestion. We have added the data related to ATRA effect on preformed biofilm as suggested (Supplementary file Figure S1). 7. Have the authors evaluated the ability of atRA to synergize with AmB or other clinically relevant antifungal compounds? This is especially relevant given the rise of resistance in Candida spp.
Authors: Dear Reviewer, thank you for pointing out this issue. A synergistic effect between ATRA and conventional antifungal drugs such as AmB and posaconazole has been already studied in our previous work ( Campione et al. 2021). In the future, the potential synergy between ATRA and antifungals currently available, i.e., AmB, posaconazole, fluconazole, will be evaluated also against drug-resistant Candida spp. strains.
Campione E et al. Antifungal Effect of All-trans Retinoic Acid against Aspergillus fumigatus In Vitro and in a Pulmonary Aspergillosis In Vivo Model. Antimicrob. Agents Chemother. 2021, 65, e01874-20. doi:10.1128/AAC.01874-20
- The authors report the inhibitory effect of atRA on yeast to hyphal transition using images. It is unclear from the methods if this was done under hyphae inducing conditions used in other papers such as Sun et al 2015, Ha et al 1999 etc. The representative images also need to be supported by quantification to support the conclusions.
Authors: Dear Reviewer, thank you for pointing this out. To induce Candida germination and hyphal growth we cultured Candida yeasts under stress conditions in RPMI 1640 supplemented with 10% fetal calf serum (FCS) at 37 °C. (Kabir MA et al, 2012) as specified in the material and methods section.The images reported in Figure 4 represent the biofilms of Candida cultures incubated for 24 hours and stained with crystal violet (CV), before CV solubilization with glacial acetic acid, for spectrophotometer biofilm analysis. Thus, the results of quantification of Candida control and ATRA-treated groups, measured as total biofilm biomass of C. albicans, have not been shown in the paragraph 3.4 because they were found to be substantially similar to those already presented in Figure 1. Thank you for this comment, which has allowed us to clarify a very important point. Kabir MA, Hussain MA, Ahmad Z. Candida albicans: A Model Organism for Studying Fungal Pathogens. ISRN Microbiol. 2012;2012:538694.
- Given the potent effects of atRA on mammalian cells, the authors should discuss the existing knowledge regarding the use of atRA as a therapeutic agent, in terms of bioavailability, safety, potential cytotoxicity etc.
Authors: Thank you for this comment. We have implemented a specific discussion on this in the revised copy of our manuscript by citing specific literature and outlining that much work is required before reaching definitive conclusions.
Thank you for your valuable comments
Best regards

Round 2
Reviewer 1 Report
The authors have greatly improved their manuscript, and have carefully addressed most of the comments raised.
However, I could not find the Figure/Images the authors report in their answer to my question on DMSO. Here the authors stated the following: "…..At this concentration DMSO did not affect Candida growth and germination in our experimental models. In this respect, we have provided the images of Candida cultures in the absence or presence of DMSO, showing that DMSO at 2.5% did not inhibit germination of Candida yeasts.” I could not find these images (maybe I am missing some files).
It would be important (given the controversial effects of DMSO discussed in published literature) to add these images to the supplementary material, stating in the material and methods section that these concentrations were tested as "safe" in your hands, for your particular readouts.